# LVCap-Eval: Towards Holistic Long-form Video Caption Evaluation for Multimodal LLMs

## Abstract

Generating coherent and factually grounded captions for long-form videos is a critical yet underexplored challenge for multimodal large language models (MLLMs). Existing benchmarks, which predominantly feature short clips, are insufficient for evaluating a model's ability to capture narrative structure and fine-grained details over extended durations. To address this gap, we introduce LVCap-Eval, a benchmark for long-form video captioning. LVCap-Eval comprises 200 videos from six diverse domains, ranging from 2 to 20 minutes, and features a dual-dimension evaluation protocol that assesses both scene-level narrative coherence and event-level factual accuracy. To facilitate model improvement, we also provide a pipeline for generating a training corpus, demonstrating that fine-tuning with as few as 7,000 samples yields substantial gains. Our evaluation of existing MLLMs on this benchmark reveals a significant performance disparity: while leading closed-source models (e.g., Gemini-2.5-Pro) perform robustly across various video durations, their open-source counterparts degrade sharply as video length increases. Finally, our analysis of these model failures highlights potential directions for improving the long-video comprehension of MLLMs.

## 1 Introduction

Video captioning, the task of generating textual descriptions of video content, serves as a foundational component for downstream applications such as video generation (Fan et al., 2025) and media analysis (Song et al., 2024). Recent advancements in data pipelines (Chen et al., 2024a), optimization techniques like direct preference optimization (Yuan et al., 2025), and grounding methods (Guo et al., 2024) have substantially improved caption quality, achieving state-of-the-art performance on short-form benchmarks like MSR-VTT (Xu et al., 2016) and YouCook2 (Zhou et al., 2018).

However, real-world applications increasingly demand tackling long-form, minute-level videos. This shift brings forth a range of complex challenges that seldom appear in short-form tasks, including constraint on context window length owing to the limited GPU memory (Rasley et al., 2020), maintaining semantic consistency (Jung et al., 2025), modeling long-term temporal dependencies (Wang et al., 2024a), segmenting scenes (Chen et al., 2021), and performing character re-identification (McLaughlin et al., 2016).

The proliferation of long-form video captioning approaches (Ma et al., 2025; Zhang et al., 2024) has enabled practical applications like video chaptering (Ventura et al., 2025), copy creation (Wu et al., 2025), and training data preparation for video generation (Henschel et al., 2025). However, current evaluation methods lag behind the advancements on long-form video understanding (Wang et al., 2025b; Chen et al., 2024b). Existing benchmarks are inadequate, as they either test for broad comprehension of extended-length videos via multi-choice question-answers (Zhou et al., 2025; Wu et al., 2024) or assess caption quality in limited contexts (Xu et al., 2025; Chai et al., 2024), thereby failing to measure narrative detail and factual accuracy.

Establishing a more effective evaluation framework requires addressing two primary challenges: (1) defining a captioning format that is both descriptively rich and useful for downstream tasks, and (2) developing robust metrics to assess the factual accuracy of these captions in diverse forms or scene segmentation.

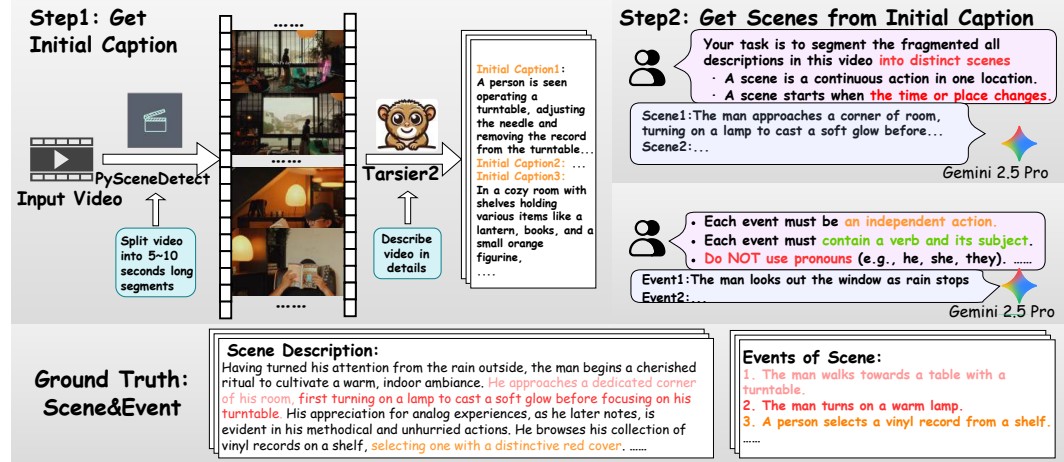

Figure 1: Overview of ground-truth generation procedure. Given each video, we first extract the video captions for each split video. Then, we extract the scenes and events and perform human validation to ensure quality.

To address the above limitations, as shown in Figure 1, we introduce LVCap-Eval, a hierarchical benchmark for evaluating the long-form video captioning capabilities of MLLMs. The benchmark consists of 200 narrative-rich videos, each 2–20 minutes in length, drawn from six distinct domains. LVCap-Eval leverages a dual-dimension evaluation protocol that assesses both **scene-level** narrative coherence and **event-level** factual accuracy. This structure enables a comprehensive analysis of a model's ability to understand both the global storyline and its underlying details. The benchmark is built upon a novel hybrid pipeline that integrates automated segmentation, dense captioning, and LLM-based structuring, with a final human validation stage to ensure both scalability and high factual fidelity. The main contributions are summarized as follows:

- **A high-quality long-form video caption benchmark with robust evaluation protocol:** We introduce LVCap-Eval, the first benchmark to systematically and robustly evaluate long-form video captioning regardless of the form of descriptions. We propose a multi-dimensional evaluation protocol, which includes a scene score to assess deep contextual reasoning and an event score to rigorously measure factual recall and precision.

- **An automatic training dataset generation pipeline:** The fine-tuned LV-Captioner-Qwen-7B illustrates that the curated datasets can significantly improve the accuracy and fidelity of the captioning performance, outperforming even larger open-source baselines.

- **Highly challenging and discriminative tasks:** Our empirical evaluation on long video captioning demonstrates the task's discriminative power, exposing a substantial performance gap between open-source models and their closed-source counterparts. While performance varies widely, even the best-performing systems achieve only moderate results. These findings indicate that generating long-form video captions warrants further investigation.

## 2 RELATED WORK

**Long Video Understanding Benchmarks and Video Caption Benchmarks.** The advancement of Video Language Models (VLMs) has shifted evaluation beyond short-clip benchmarks like MSR-VTT and YouCook2 (Xu et al., 2016; Zhou et al., 2018), whose limited durations constrain narrative reasoning. Early work such as MovieQA and TVQA (Tapaswi et al., 2016; Lei et al., 2018) introduced narrative QA for films and TV shows, paving the way for longer contexts. Recent benchmarks extend both scale and task coverage: Video-MME examines performance under increasing duration (Fu et al., 2024); MLVU handles hour-level multi-task reasoning (Zhou et al., 2025); and LongVideoBench targets long-range tracking with interleaved video–subtitle inputs (Wu et al., 2024). Beyond QA, long-form captioning benchmarks evaluate descriptive ability at extended

timescales (Xu et al., 2025; Chai et al., 2024; Liu et al., 2025; Tang et al., 2024). Large-scale corpora like VATEX and NoCaps expand open-domain coverage (Wang et al., 2019; Agrawal et al., 2019), while DetailCaps and CompreCap emphasize fine-grained faithfulness (Dong et al., 2024; Lu et al., 2025). These resources not only test the limits of temporal reasoning but also encourage models to balance narrative coherence, factual precision, and domain generalization. The above-mentioned benchmarks have focused solely on answering questions about one or several clips from long videos, or measuring the fidelity of descriptions of short clips. LVCap-Eval combines these two aspects to align with the demands of industrial-level video captioning applications, offering a more holistic evaluation paradigm for practical deployment scenarios.

**Video Caption Models.** Video captioning translates visual-temporal information into natural language. Traditional evaluations have relied heavily on n-gram overlap metrics such as BLEU (Papineni et al., 2002) and CIDEr (Vedantam et al., 2015), though these often fail to capture semantic fidelity in long narratives. To address this, semantics-aware metrics like SPICE and BERTScore (Anderson et al., 2016; Zhang et al., 2019), as well as multimodal metrics like CLIPScore (Hessel et al., 2021), have been introduced. Recent advances in proprietary systems (e.g., Gemini, GPT-4o (Team, 2025; Google DeepMind, 2025; Hurst et al., 2024)) and open-source VLMs (e.g., Qwen2.5-VL (Bai et al., 2025), InternVL3.5 (Zhu et al., 2025; Wang et al., 2025a), MiniCPM-V 4.5 (OpenBMB Team, 2025)) demonstrate strong performance on complex video understanding. Specialized captioning models such as the Tarsier series (Yuan et al., 2025; Wang et al., 2024b) focus on fine-grained short-video description, but extending this to long-form captioning remains an open challenge. LVCap-Eval provides a stable and discriminative assessment of the model's capabilities in long-form video captioning, and can enlighten the model's future development.

## 3 LVCap-Eval

### 3.1 Overview

LVCap-Eval is proposed to address the critical challenge of evaluating MLLMs for long-form video captioning. As shown in Table 1, LVCap-Eval comprises 200 long-form videos (2–20 minutes), spanning six major categories and multiple sub-categories to ensure diversity. In addition, the benchmark is carefully designed to cover a wide range of real-world scenarios, thereby reflecting the practical challenges faced in industrial applications.

Besides, LVCap-Eval introduces a dual-dimension evaluation framework: a coarse-grained scene dimension for holistic narrative understanding, and a fine-grained event dimension for capturing specific details and their temporal evolution. This dual perspective enables comprehensive measurement of both global coherence and local accuracy, offering richer diagnostic signals for model development.

### 3.2 Video Collection

We construct the benchmark from 200 manually selected videos from YouTube. To ensure timeliness and diversity, the video collection consists of English and Chinese content published within the last two years, with durations of 2 to 20 minutes. The dataset is structured into six primary categories, each containing 30 videos. As detailed in Figure 2, each primary category is composed of several distinct sub-genres, reflecting heterogeneous narrative structures. Furthermore, the dataset is augmented with 20 additional videos from other genres, including news reports and interviews, to increase the variety.

### 3.3 Annotation Pipeline

We adopt a hierarchical annotation pipeline comprising three automated steps, followed by a final stage of human validation. Initially, PySceneDetect (Castellano & the PySceneDetect contributors, 2025) is used to segment each video into shots, typically ranging from 5 to 10 seconds in length, while preserving original timestamps. Subsequently, each shot is processed using Tarsier2 (Yuan et al., 2025), which generates dense and factual captions. These captions are further refined by Gemini-2.5-Pro, which is prompted with a detailed definition of scene in Appendix B. Gemini-2.5-Pro organizes the captions into distinct scenes based on their content. Finally, to enable fine-grained

Table 1: Comparison of different video benchmarks.

| Benchmark | Evaluation | Main Duration | Source |
|---|---|---|---|
| LongVideoBench (Wu et al., 2024) | QA on long videos | 8s–60m | Life, Movie, Knowledge, News |
| Video-MME (Fu et al., 2024) | Multi-task QA | 1s–60m | Life, Movie/TV, Sports, Knowledge |
| MLVU (Zhou et al., 2025) | QA across domains | 3m–120m | Life, Movie/TV, Sports, Knowledge, Simulation |
| CaReBench (Xu et al., 2025) | Fine-grained caption | 5s–30s | Personal care, Socializing, Sports, Household |
| VDC (Chai et al., 2024) | Structured caption | 10s–60s | Nature, Film, Technology |
| CAPability (Liu et al., 2025) | Multi-dim caption | 1s-434s | Environments,Entertainment, Artistic Content |
| Video-SALMONN2 (Tang et al., 2024) | Caption quality | 30s–60s | Entertainment, News, Sports, Lifestyle |
| **LVCap-Eval (Ours)** | **Caption on scenes and events** | **2m–20m** | **Six primary categories of long-form video** |

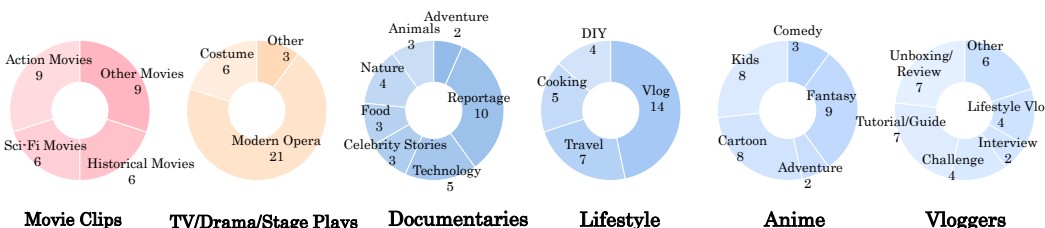

Figure 2: Statistics over the primary categories of videos in LVCap-Eval.

factuality analysis, we additionally adopt a event extraction pipeline inspired by the DREAM-1K benchmark (Wang et al., 2024b), which extracts events from each scene. Finally, human annotators review the results to ensure accuracy by verifying scene boundaries, validating the correctness of each scene and event.Our ground truth consists of the aforementioned events and scenes, and the statistical information for the ground truth of all samples is shown in the Figure 3.

### 3.4 EVALUATION PROTOCOL

As illustrated in Figure 4, we propose a novel dual-dimensional evaluation framework and detail its evaluation pipeline at both the scene and event levels. This framework is designed to simultaneously assess a model's narrative comprehension and factual accuracy, thereby providing a comprehensive evaluation of the performance of various mainstream open-source and proprietary large models on the long-video captioning task.

**Scene-level Evaluation:** Based on the final scene descriptions obtained from the annotation process described in Section 3.3, we automatically generate a structured set of 10 question–answer pairs for each scene. Specifically, the scene descriptions and the prompt in Appendix B are provided to Gemini-2.5-Pro to generate 10 structured question–answer pairs for the current scene. These pairs cover evaluations across eight distinct dimensions: temporal, atmosphere, emotion, causality, relations, cinematic, symbolism, and factuality. All questions are yes/no questions.

The tested model first produces a single, scene-level description for the entire video. An evaluation LLM then reads this description together with each question and the ground truth reference points, giving a correct/incorrect judgment. The tested model generates a description for the video, which is then paired with corresponding questions and evaluated using GPT-4.1(OpenAI, 2025). GPT-4.1 answers each question based on the provided description and determines the correctness of each response. The scene-level accuracy score, denoted as $S_s$, is defined as the overall accuracy across

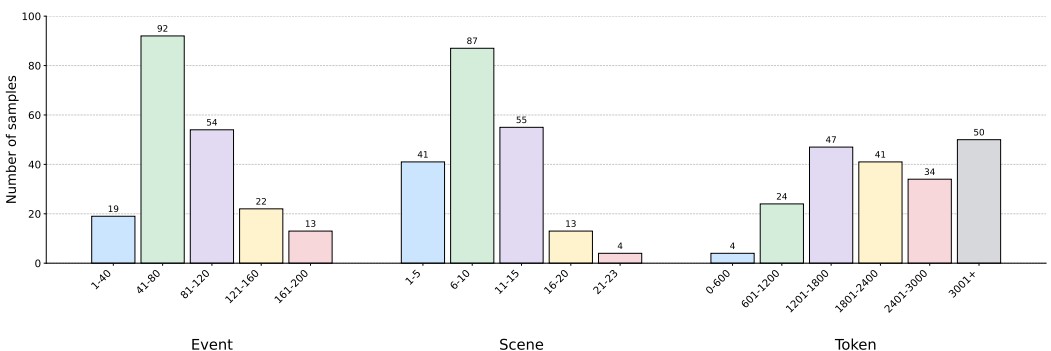

Figure 3: Distributions of events, scenes, and tokens.

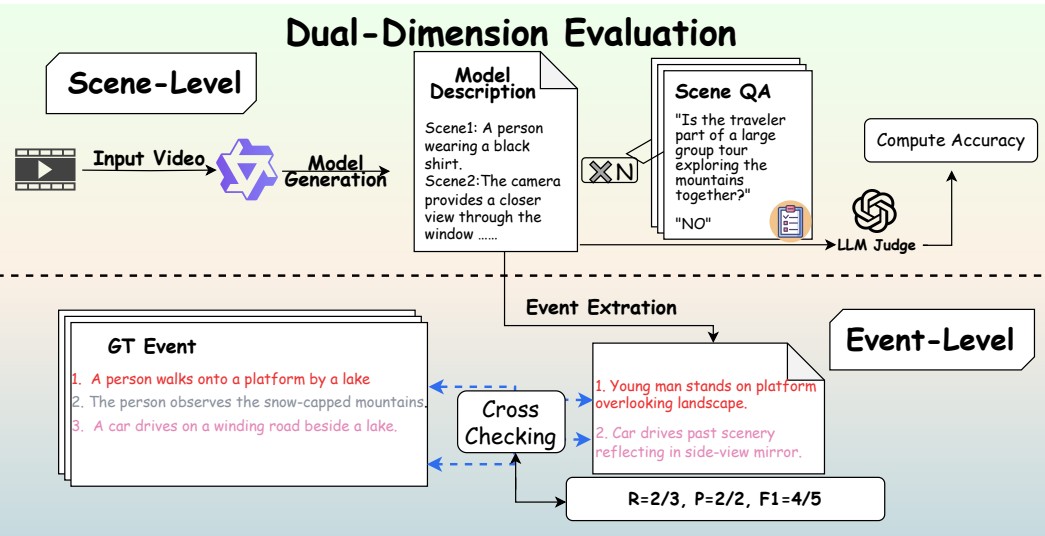

Figure 4: Overview of the evaluation process. For each input video, we first generate the scene descriptions and then extract the events from these descriptions. Finally, we compute the scene-level and event-level scores based on the prediction and ground truth.

the question set. Specifically, let $|Q|$ represent the total number of questions and $|Q^{\checkmark}|$ the number of correct responses; $S_s$ is computed as $S_s = \frac{|Q^{\checkmark}|}{|Q|} \times 100\%$.

**Event-level Evaluation:** To capture fine-grained factuality, we adopt a two-step pipeline inspired by the DREAM-1K benchmark (Wang et al., 2024b), which combines event extraction and textual entailment.

First, given the reference description $D_{\text{ref}}$ and the model-generated description $D_{\text{model}}$, we use GPT-4.1(OpenAI, 2025) as the extractor to derive event sequences $E_{\text{ref}}$ and $E_{\text{model}}$. Second, we utilize GPT-4.1 as the judge to calculate the number of matched events between $E_{\text{ref}}$ and $E_{\text{model}}$. Based on these matches, we further compute recall ($R$) and precision ($P$):

$$R = \frac{|\{\, e \in E_{\text{ref}} \mid D_{\text{model}} \vDash e \,\}|}{|E_{\text{ref}}|}, \quad P = \frac{|\{\, e \in E_{\text{model}} \mid D_{\text{ref}} \vDash e \,\}|}{|E_{\text{model}}|}, \quad F1 = \frac{2PR}{P+R}. \quad (1)$$

We use $S_e$ to represent the event-level score $S_e = F1 \times 100\%$ .

## 3.5 TRAINING SET

To advance research in long-form video understanding, we have constructed a new training dataset comprising 7,000 videos. This dataset originates from an initial collection of 2,000 long-form videos, each ranging from 3 to 40 minutes in duration, sourced from a diverse array of public benchmarks, including ActivityNet (Caba Heilbron et al., 2015), YouCook2 (Zhou et al., 2018), MM-Trail (Chi et al., 2024), YouTube-Commons (PleIAs, 2024), Omega-Multimodal (OMEGA Labs, Inc., 2024), and HD-VILA-100M (Xue et al., 2022).

To enhance the temporal diversity and volume of our data, we apply a dual-segmentation strategy to each source video: (1) first, dividing each video into two equal halves, and (2) subsequently, partitioning each half into two segments of random duration. After excluding clips that fall below a predefined duration threshold, we obtain the final curated dataset of 7,000 videos. Each video is then annotated with fine-grained captions, leveraging the same automated annotation pipeline developed for our LVCap-Eval benchmark, ensuring consistency in quality. Furthermore, to generate coarse-grained scene annotations, we employ GPT-4-mini (OpenAI, 2024) to programmatically merge consecutive, thematically related fine-grained captions. This process effectively synthesizes detailed event descriptions into a cohesive scene .

## 4 EXPERIMENTS

### 4.1 EXPERIMENTAL SETTINGS

We evaluate a comprehensive suite of 18 state-of-the-art MLLMs on LVCap-Eval. Both leading closed-source models and a wide spectrum of open-source alternatives are involved to ensure a diverse and fair comparison. The evaluated models include: **Closed-source:** Doubao-seed-1-6-vision (ByteDance Seed Team, 2025),Gemini-2.0-Flash and Gemini-2.5-Pro (Google DeepMind, 2025). **Open-source:** InternVL3 (Zhu et al., 2025), the InternVL3.5 series (Wang et al., 2025a) , LongVILA (OpenBMB Team, 2025), the Qwen2.5-VL series (Bai et al., 2025) Tarsier2 (Yuan et al., 2025), TinyLLaVA-Video (Zhang et al., 2025b), VideoChat-Flash-Qwen2-7B (Li et al., 2024), and VideoLLaMA3 (Zhang et al., 2025a).

### 4.2 MAIN RESULTS

The performance of all models is summarized in Table 2, with the following key findings:

**Closed-source models establish a dominant baseline, far surpassing open-source counterparts.** We create a simple visualization of Table 2, as shown in Figure 5. Gemini-2.5-Pro achieves state-of-the-art performance, leading the benchmark by a significant margin and highlighting the capability gap between proprietary and open-source models. While larger model variants within families (e.g., InternVL) show improved results due to scaling effects, their performance remains inferior to closed-source models. However, overall scores across all models suggest that long-form video captioning remains a challenging task. Additionally, the inclusion of reasoning steps offers only marginal improvements.

**Open-source models suffer severe performance degradation with increasing video duration.** The scores of most open-source models gradually decrease as video duration increases. For some models, such as MiniCPM-V 4.5-thinking, a significant drop in performance occurs after exceeding certain time brackets. For example, its Scene score drops by 9.79 points when transitioning from the 6–10 minute segment to the 16–20 minute one. In contrast, Gemini-2.5-Pro shows remarkable stability, with its scores varying by only 1.6 points across all durations, indicating a critical advantage of closed-source models.

**Fine-tuning with the proposed dataset significantly improves captioning performance for long-form videos.** Fine-tuning Qwen2.5-VL-7B using Low-Rank Adaptation (LoRA) (Hu et al., 2022) yields the derived model **LV-Captioner-Qwen-7B**, which not only surpasses its base model but also outperforms the larger Qwen2.5-VL-32B. Importantly, LV-Captioner-Qwen-7B demonstrates substantial gains in the most challenging video segments (11–20 minutes), validating the efficacy of the proposed training set construction pipeline in enhancing long-form captioning quality.

Table 2: Comparison of 18 MLLMs and Ours on LVCap-Eval.We use blue to show the best score in the current group.

| Model | Overall Performance | | Performance by Video Duration (min) | | | | | | | |
|---|---|---|---|---|---|---|---|---|---|---|
| | | | 2-5 mins | | 6-10 mins | | 11-15 mins | | 16-20 mins | |
| | $S_s$ | $S_e$ | $S_s$ | $S_e$ | $S_s$ | $S_e$ | $S_s$ | $S_e$ | $S_s$ | $S_e$ |
| *Closed-Source Models* | | | | | | | | | | |
| Gemini-2.5-Pro | 57.13 | 60.69 | 55.73 | 61.12 | 57.24 | 60.43 | 57.33 | 60.76 | 57.28 | 61.92 |
| Doubao-Seed-1-6-vision | 53.59 | 59.71 | 51.61 | 60.35 | 55.90 | 59.75 | 53.16 | 57.41 | 51.68 | 58.48 |
| Gemini-2.0-Flash | 43.85 | 53.01 | 42.07 | 52.33 | 45.71 | 55.91 | 43.47 | 51.78 | 41.99 | 52.24 |
| *Open-Source Models (Thinking)* | | | | | | | | | | |
| MiniCPM-V 4.5-thinking | 41.77 | 44.42 | 45.97 | 55.16 | 45.35 | 45.89 | 36.45 | 39.52 | 35.56 | 32.67 |
| InternVL3.5-38B-thinking | 37.92 | 42.66 | 41.96 | 49.04 | 39.09 | 42.53 | 33.37 | 38.88 | 36.37 | 39.11 |
| InternVL3.5-14B-thinking | 36.46 | 38.29 | 40.43 | 42.92 | 38.38 | 40.35 | 33.02 | 35.73 | 32.52 | 35.18 |
| *Open-Source Models (>10B)* | | | | | | | | | | |
| Qwen2.5-VL-72B | 38.71 | 42.13 | 40.33 | 49.38 | 42.99 | 42.75 | 34.28 | 38.21 | 32.37 | 39.46 |
| InternVL3.5-38B | 37.50 | 40.17 | 40.45 | 43.71 | 40.94 | 42.64 | 31.64 | 38.07 | 31.37 | 35.59 |
| InternVL3.5-14B | 34.09 | 36.48 | 37.22 | 41.19 | 36.78 | 36.83 | 30.09 | 33.61 | 28.12 | 30.22 |
| Qwen2.5-VL-32B | 32.35 | 38.37 | 34.54 | 45.03 | 31.97 | 38.66 | 31.98 | 36.61 | 28.74 | 31.88 |
| *Open-Source Models (<10B)* | | | | | | | | | | |
| LV-Captioner-Qwen-7B (Ours) | 35.71 | 38.23 | 36.82 | 42.41 | 36.02 | 38.91 | 35.96 | 36.26 | 34.47 | 33.84 |
| InternVL3.5-8B | 32.09 | 32.55 | 37.91 | 38.44 | 33.39 | 33.12 | 26.48 | 26.79 | 27.87 | 31.37 |
| Qwen2.5-VL-7B | 29.59 | 33.86 | 31.41 | 38.58 | 30.06 | 34.05 | 28.16 | 32.91 | 26.30 | 29.49 |
| Tarsier2-7B | 26.71 | 23.31 | 36.54 | 27.56 | 24.95 | 24.13 | 25.92 | 22.84 | 20.09 | 16.43 |
| InternVL3-8B | 24.56 | 26.77 | 29.75 | 28.23 | 24.72 | 25.98 | 21.69 | 24.41 | 21.90 | 21.57 |
| VideoLLaMA3-7B | 24.03 | 25.68 | 34.41 | 31.02 | 24.24 | 27.46 | 20.92 | 19.53 | 14.19 | 18.89 |
| VideoChat-Flash-Qwen2-7B | 23.51 | 24.14 | 29.14 | 28.78 | 23.87 | 27.26 | 20.68 | 20.97 | 20.28 | 17.65 |
| TinyLLaVA-Video | 19.79 | 12.82 | 15.53 | 14.39 | 18.04 | 12.06 | 17.75 | 12.44 | 10.24 | 8.71 |
| LongVILA | 16.46 | 20.25 | 18.57 | 21.69 | 18.92 | 22.33 | 12.68 | 19.08 | 13.49 | 14.96 |

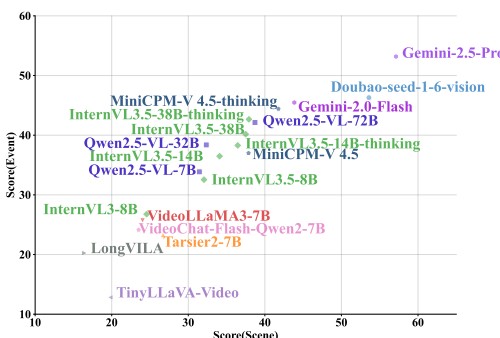

Figure 5: Main results of 18 MLLM.

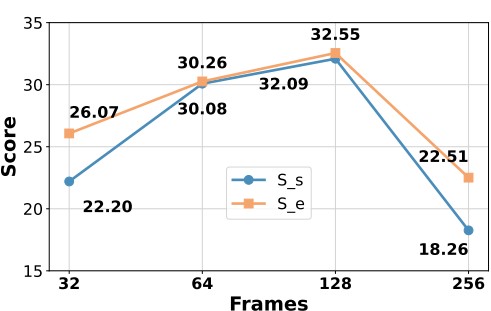

Figure 6: Score of different frames.

## 4.3 FURTHER ANALYSIS

**Effect of Input Frames.** To investigate the relationship between visual information density and task performance, we systematically varied the number of sampled input frames for the InternVL-3.5-8B model, ranging from 32 to 256. The results, as depicted in Figure 6, reveal a distinct trend: model performance progressively improves up to an inflection point at 128 frames, after which it begins to degrade. This non-linear relationship suggests an optimal trade-off between acquiring sufficient visual information and overwhelming the model's finite contextual capacity. Beyond this threshold, an excessive volume of visual tokens appears to saturate the context window, displacing the very attentional resources required for high-level reasoning and narrative synthesis.

This shows us something very important: to be good at describing long videos, just adding more frames is not enough. It's true that more frames can help the model see more, up to a point. But it is just as important for the model to be able to "think" about the connections between different frames over time. A bigger context window helps with both problems: it lets the model see more frames and also leaves more room for it to reason about what it sees.

**Evaluation on Different Video Categories.** To assess model robustness across diverse video genres, we compare Gemini-2.5-Pro and Qwen-2.5VL-7B on the six categories: Movie Clips, TV/Drama/Stage Plays, Documentaries, Lifestyle, Anime, and Vlogs. As detailed in Figure 7, the results reveal a significant performance disparity, with Gemini-2.5-Pro consistently outperforming Qwen-2.5VL-7B. Although Gemini-2.5-Pro's performance is relatively stable, it shows a vulnerability to complex temporal dependencies, achieving its lowest Scene score of 53 on Vlogs. Conversely, Qwen-2.5VL-7B exhibits pronounced difficulty with genres rich in human interaction and dynamic narratives, evidenced by its low Scene score of 21 on Movie Clips. These findings indicate that performance is highly genre-dependent and that processing complex, event-driven narratives remains a primary challenge for state-of-the-art models.

**Error Analysis.** An error analysis of Gemini-2.5-Pro and Qwen2.5-VL-7B across eight question categories reveals divergent performance profiles (Figure 8). The closed-source Gemini-2.5-Pro demonstrates remarkably consistent performance, with scores across all categories falling within a narrow range of [55.31, 59.61]. This stability suggests a robust and well-rounded generalization capability, allowing it to handle diverse question types with similar proficiency. In stark contrast, the open-source Qwen2.5-VL-7B exhibits distinct vulnerabilities. Its accuracy drops sharply on categories requiring abstract reasoning (e.g., Symbolism, 23.09) and fine-grained factual recall (e.g., Accuracy, 25.09). This disparity strongly indicates that while the open-source model may be competent in general tasks, its primary limitations lie in higher-order cognitive abilities, such as interpreting abstract concepts and retaining precise details.

**Evaluation on Different Numbers of Scenes and Events.** In this experiment, we investigate the impact of varying numbers of scenes and events in the ground truth on the performance of the InternVL3-8B model. The results, presented in Figure 9 and Figure 10, reveal a clear decreasing trend in both scene scores ($S_s$) and event scores ($S_e$) as the number of scenes or events increases. Specifically, when the number of scenes in the ground truth rises, the scene scores gradually drop from 39.38 to 16.04. Similarly, increasing the number of events leads to a decline in event scores, from 42.06 to 17.68. This suggests that as more distinct scenes and events are introduced, the video's thematic complexity increases, making it more difficult for the model to track and accurately describe all relevant information. Consequently, handling videos with rich and varied narrative structures remains a significant challenge for the model.

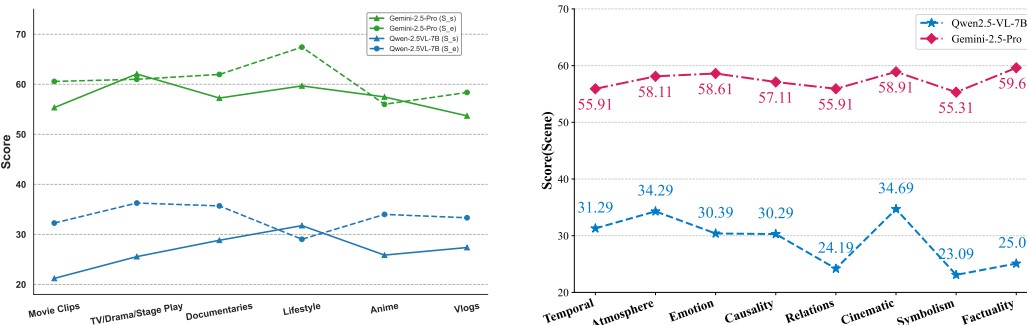

Figure 7: Score of different video types.    Figure 8: Score of different question types.

**Different Evaluation Methods: (1) Isolating Perception and Reasoning Bottlenecks.** To precisely isolate the performance bottleneck between visual perception and narrative reasoning, we designed and conducted a hybrid evaluation. In this setup, we first segment source videos into standardized 10-second clips, following established benchmarks like Kinetics (Kay et al., 2017) to

ensure comparability. We then employ a single, powerful vision-language model (Qwen-2.5VL-7B) to function as a unified perceptual front-end. This model generates a descriptive caption for each clip, effectively standardizing the visual-to-text conversion and holding the perceptual component constant across the experiment. These fixed, clip-level descriptions are then fed as common input to two distinct reasoning modules: the original model's own text-based reasoner and that of a more advanced, text-only model (Gemini-2.5-Pro). As detailed in Table 3, by replacing the model's native reasoning module with the superior text-only reasoner, we observed a dramatic improvement in the Scene score, which surged from 20.05 to 42.22. This striking outcome strongly indicates that the primary performance limitation in current MLLMs does not lie in fundamental visual perception, but rather in the high-level cognitive task of narrative reasoning. This suggests that for some open-source models, their underperformance may stem from an inability to synthesize information from frames into a coherent and comprehensive description.

**Different Evaluation Methods: (2) Context-aware Segmental Evaluation.** To investigate performance degradation on longer videos, we introduce a context-aware segmental evaluation protocol. In this protocol, we partition videos by ground truth scene boundaries and task the model (Qwen-2.5VL-7B) with describing each segment sequentially. Crucially, the caption for each segment is conditioned on the captions generated for the preceding ones, simulating a simple memory mechanism. As shown in Table 4, this approach yields a modest but clear improvement in narrative understanding, increasing the Scene score from 29.59 to 31.45 while the Event score remains stable. These results suggest that even a basic sequential context-passing mechanism can help mitigate challenges posed by long-range dependencies.

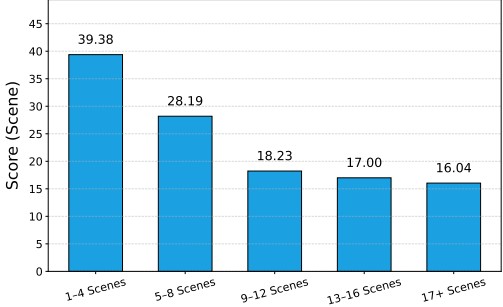

Figure 9: Score by number of scenes.      Figure 10: Score by number of events.

Table 3: Performance with different narrative reasoners.

| Reasoner | $S_s$ | $S_e$ |
| --- | --- | --- |
| Qwen-2.5VL-7B | 20.05 | 23.15 |
| **Gemini-2.5-Pro** | **42.22** | **43.39** |

Table 4: Performance with or without provided scene boundaries.

| Model | $S_s$ | $S_e$ |
| --- | --- | --- |
| Qwen-2.5VL-7B (Baseline) | 29.59 | **33.86** |
| Qwen-2.5VL-7B (Segmental) | **31.45** | 33.42 |

## 5 CONCLUSION

This paper presents LVCap-Eval, a high-quality benchmark specifically designed for the evaluation of long video captioning systems. By employing a novel LLM-driven annotation pipeline and a dual-dimension evaluation protocol, LVCap-Eval assesses two critical aspects of captioning quality: narrative coherence and fine-grained factual accuracy. Extensive evaluations reveal a significant performance gap among models, with open-source models performing worse than closed-source ones. These findings highlight the inherent complexity of long video captioning and further indicate that there are still numerous challenges to overcome in using large language models (LLMs) for generating captions for long videos. Specifically, key issues include processing larger frame counts, improving context length, and reasoning about information between frames. Addressing these aspects demands innovation in model architecture, training strategies, and dataset design.

## ETHICS STATEMENT

LVCap-Eval is released solely for academic research purposes under the Creative Commons Attribution-NonCommercial 4.0 International License (CC BY-NC 4.0) . All videos remain the copyright of their respective owners. This license permits sharing and adaptation of the dataset for academic and non-commercial purposes, provided appropriate credit is given. Any form of commercial use, distribution, publication, copying, dissemination, or modification of LVCap-Eval, in whole or in part, without prior approval, is strictly prohibited.

This research has been conducted in full compliance with relevant laws, regulations, and ethical standards. No unethical practices were involved during the data collection and processing. No individuals were coerced or forced to annotate data; all participation was voluntary and respectful of participants' rights.

## REPRODUCIBILITY STATEMENT

We will make all resources and code publicly available. Our well-defined protocols ensure stable and reliable evaluation results. We strive to ensure the reproducibility of our research. All datasets used in this study are either publicly available or can be shared upon reasonable request.

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

## A  THE USE OF LARGE LANGUAGE MODELS

In preparing this paper, we used large language models (LLMs) in two ways. First, LLMs were employed to aid and polish the writing. Second, LLMs were used for retrieval and discovery, such as finding related work. Importantly, the research design, experimental results, and analysis were not generated or influenced by LLMs. All scientific contributions are entirely the work of the authors.

## B  PROMPTS

---

**Prompt1:Get Ground Truth**

ROLE AND GOAL

You are an intelligent **scene segmenter** and **narrative generator**. Your task is to process video transcripts, segment them into coherent scenes, and generate a detailed narrative description for each scene.

SCENE DEFINITION FOR SEGMENTATION

A scene is a cohesive dramatic action unit occurring in a continuous time period and a single physical location. A new scene begins when there is a significant change in time or location that disrupts narrative continuity.

PROCESSING INSTRUCTIONS

1. **Incremental Scene Segmentation**: Analyze each incoming sentence for signals indicating a scene boundary. These signals include:
   - *Spatial/Temporal Shift*: The narrative explicitly mentions a different physical location or jumps forward/backward in time.
   - *Thematic/Objective Shift*: The primary goal, motivation, or emotional tone of the characters fundamentally changes.

2. **Cohesive Scenes**: Avoid generating too many granular scenes. A scene should represent a substantial and meaningful narrative block. If no significant boundary signal is detected, add the sentence to the current scene's context.

OUTPUT INSTRUCTIONS

For each completed scene, you must generate a single, detailed narrative description. This description must adhere to the following rules:

• **Length and Style**: It must be a detailed, engaging paragraph of at least 150 words, written from a third-person objective point of view.

• **Core Elements**: It must incorporate the six core storytelling elements: Who (characters), Where (setting), When (time), What (action), Why (motivations), and the emotional tone.

• **Vivid Language**: Use descriptive language and the "show, don't tell" principle to create a rich and immersive narrative.

OUTPUT FORMAT

Your response must be in **JSON Lines (`.jsonl`) format**. Each line must be a separate, self-contained JSON object.

• Each JSON object represents one scene.

• It must contain a single key-value pair, where the key is a **unique sequential number as a string** (e.g., "1", "2", "3"...) and the value is the **detailed narrative description of that scene**.

• Only output JSONL content, do not output any other useless content.

---

**Example Output:**
{scene1: "A detailed narrative description of the first scene, at least 150 words long, covering who, what, where, when, and why...",}
{scene2: ......}
{scene3: ......}
Start analysis on the following captions:

## Prompt2:Extract Events

You are an AI assistant specialized in **event extraction**. Your task is to process descriptive text and extract a structured list of atomic, action-oriented events based on a comprehensive set of rules.

INPUT HANDLING INSTRUCTIONS

**IMPORTANT**: The input text you receive may contain meta-commentary, self-corrections, or a "chain of thought" from another AI. **You must ignore these sections.** Your sole focus is to identify and process the actual narrative or scene descriptions provided in the input.

EVENT EXTRACTION INSTRUCTIONS

From the given descriptive text, extract a list of concise, action-oriented events. Each event must meet the following comprehensive guidelines:

1. **Atomic Structure:**
   - Each event must be an independent, single action or movement.
   - It should not include static information or overly descriptive details.
   - Events must be atomic, meaning they cannot be split into multiple actions.

2. **Language Requirements:**
   - Each event must be written as a brief sentence with a maximum of **12 words**.
   - The sentence must include a **subject** and a **verb** (predicate), and optionally an **object**.
   - **Do NOT use pronouns** (e.g., "he," "she," "they"). Replace pronouns with the actual nouns they refer to (e.g., "the man," "Jerry," "the agent").

3. **Content Guidelines:**
   - Extract only **actions, motions, or movements** from the text.
   - Ignore all scene titles or non-descriptive headers. Focus exclusively on the descriptions of actions.

4. **Redundancy and Variation Handling:**
   - *Input Redundancy*: The source description may contain repetitive or verbose sentences. You must identify and **disregard these redundancies**. Do not extract events from these repetitive parts.
   - *Output Uniqueness*: Ensure that no two extracted events are identical or nearly identical. Aim for distinction and variation across all events.

5. **Output Format:**
   - Your response must be in **JSON Lines (`.jsonl`) format**. This means each line of the output is a separate, self-contained JSON object.
   - Each event must be a **dictionary** with a unique **sequential number as a string key** (e.g., "1", "2", "3"...) and the **event description as the string value**.

EXAMPLE OUTPUT

```
{"1": "Tom pulls Jerry out with a rope"}
{"2": "Tom washes Jerry with a bar of soap"}
```

```
{"3": "Jerry  escapes  through  a  mouse  hole"}
{"4": "Jerry  drags  the  rope  through  the  hole"}
```

---

**Prompt3:Get QA from Ground Truth**

ROLE AND GOAL

You are tasked as a **Video Description Evaluation Expert** and a **Text Quality Detection Specialist**. Your main responsibility is to design a set of Yes/No questions aimed at evaluating the quality of video descriptions produced by AI models. These questions will strictly adhere to the provided standard description" as a factual reference.

STRATEGIC OBJECTIVE

Break down the given "standard description"into an evaluation questionnaire. This evaluation tool should rigorously test for the following potential issues:

1. *Information Omission:* Check if essential details are missing.

2. *Vague Details:* Ensure descriptions are specific and clear.

3. *Logical Errors:* Identify inconsistencies or false cause-effect relations.

4. *Temporal Errors:* Detect reversals in event sequences.

5. *Lack of Emotion:* Evaluate if descriptions capture the intended emotional tone.

6. *Neglect of Symbolism:* See if deeper symbolic meanings are ignored.

OPERATIONAL MANDATE

Question design rules:

1. Strictly rely on the "standard description." Questions must correspond directly to factual content within the description, either directly or indirectly.

2. Test common flaws:

   (a) **Factual Discrepancies:** Design questions with partially correct details but subtle inaccuracies.

   (b) **Detail Errors:** Focus on overlooked details such as colors, clothing, etc.

   (c) **Logical/Causal Errors:** Frame questions with flawed cause-effect relations.

   (d) **Temporal Errors:** Reverse the event timeline in the question.

   (e) **Emotional Contradictions:** Include statements that misrepresent the emotional tone.

   (f) **Filming Techniques:** Test understanding of camera work and techniques.

3. Balance between Yes/No answers: Maintain a 1:1 ratio across all questions.

4. Avoid simple or repetitive questions. Ensure difficulties are appropriately balanced.

QUESTION-SETTING RULES

- **Rule 1:** Design 10 questions for each scene, covering 10 evaluation dimensions. If there are 13 scenes, a total of 130 questions must be created.

- **Rule 2:** For each scene, provide the full "standard description" for context and develop questions specific to that scene.

- **Rule 3:** Questions should target specific errors, with detailed and reasoned analysis for "No" answers.

ASSESSMENT DIMENSIONS

Questions should align with these dimensions:

1. **Temporal:** Verify the order of events based on the standard description's sequence.

2. **Atmosphere:** Critique the accuracy of setting details and emotional tone.

3. **Emotion:** Evaluate the understanding of characters' internal emotional states.

4. **Causality:** Test cause-effect consistency.

5. **Relations:** Explore interaction accuracy between characters.

6. **Cinematic:** Check for understanding of camera work and framing techniques.

7. **Symbolism :** Analyze the interpretation of deeper meanings tied to objects or events.

8. **Factuality:** Assess details such as clothing, color, number, etc.

OUTPUT FORMAT

Questions should be formatted as follows:

```
Scene: [Scene Number]
Description: [Standard Description of the Scene]
Question 1:
- Dimension: [Accuracy]
- Question: [A clear Yes/No question]
- Options: ["Yes", "No"]
- Correct Answer: "Yes" or "No"
- Analysis: [If "No," explain the reason]
Question 2:
- Dimension: ...
- Question: ...
- Options: ...
- Correct Answer: ...
```

**Prompt4:Get Mark for Scene**

You are a **professional judge**. Your goal is to evaluate the video descriptions provided by the tested model based on the questions and answers from the `qa_json` file. You will receive two JSON files:

- One JSON file containing QA questions and answers, named like `x_qa.json`.
- Another JSON file with the tested model's descriptions of the video content.

What you need to do:

(1) For each scene in a video, attempt to find the answer to the question in the tested model's description.

- If you find the answer and it is generally consistent with the question, respond with `yes`.
- If you do not find it or the model's description is incorrect, respond with `no`.

(2) Compare the tested model's answers with the correct answers and record the number of correct answers for each scene.

- Use an array, such as `[7,6,7,7,8]`, to record the number of correct answers. Each entry in the array corresponds to the number of correct answers for a scene.
- The array should have as many entries as there are scenes in the original question set.

(3) Calculate the percentage of correct answers for all questions in the video. The overall score for the video is 100 times the percentage of correct answers.

(4) Retain two decimal places in all score calculations.

**Example:**

- If you record `[7,6,7,7,8]`, the total score should be:

$$((7+6+7+7+8)/50) \ * \ 100\% \ = \ 70\%,$$

and it should be reported as: `total_score:  70.00`.

IMPORTANT NOTES

(1) **Retain two decimal places when calculating the score.** You must calculate the number of correct answers for all scenes before scoring. Verify your calculations internally, but do not show the verification in your response.

(2) When answering questions based on the tested model's description, ignore the scene indicator from the model and focus solely on the text description.

(3) Answers must match the model's similar descriptions nearby in the "scene description" of each scene from the input QA.

(4) The `model_name` must be: `gemini-2.5-pro.jsonl`.

(5) Your evaluation output must be in **JSONL format**, including:

- Test model name.
- Video ID.
- Number of correct questions per scene.
- Total score.

(6) Ensure the output format is strictly as follows:

{"model": "xxx", "video_id": i, "scores": [10,9,6,4,5,6,7,9], "total_score": 70.00}

## B.1 PROMPT5

**Prompt5:Get Mark for Event**

You are a professional **event-matching evaluator**. Your task is to assess the accuracy of the events generated by a tested model by comparing them with ground truth (GT) event lists. You will receive two event lists:

1. **Ground Truth (GT)**: A reference list of events extracted through human annotation.

2. **Model Output**: The list of events generated by the tested model for the same captions.

Your objective is to perform event matching, calculate recall, precision, and F1 score, and output the evaluation results in JSONL format strictly following the guidelines below.

GUIDELINES

**Matching Criteria**:

• As long as the characters roughly match, it can be considered a successful match. For example, if the ground truth mentions that "a certain woman did xxx" and the tested description states "Taylor did xxx," it is acceptable as long as the action "xxx" roughly aligns.

• Match events based on meaning and logical correctness rather than superficial wording.

• Synonyms, paraphrases, and variations in phrasing are considered valid matches.

• Ignore grammatical details or minor spelling differences.

• Rough consistency in the main subject and action is sufficient for a successful match. Events classified as 'entailment' are also considered a successful match. For example, if the prediction states: 'Xiao Ming runs up and down the stairs repeatedly', and the GT states: 'Xiao Ming runs up the stairs', this should also be considered a successful match.

**Metrics Calculation**:

• **Recall**: The proportion of GT events that are correctly matched by the model's output. Formula: `Recall = (Number of Matched GT Events) / (Total Number of GT Events)`.

• **Precision**: The proportion of model-generated events that correctly match GT events. Formula: `Precision = (Number of Matched GT Events) / (Total Number of Model Events)`.

• **F1 Score**: The harmonic mean of recall and precision. Formula: `F1 = 2 * (Recall * Precision) / (Recall + Precision)` (round the final result to two decimal places).

**Simplified Output**:

• Simply output the following fields in **JSONL format**, with all numbers keeping two digits after the decimal point:

  – `"video_id"`: The ID of the video being evaluated.
  – `"model_name"`: The name of the tested model (use "merge").
  – `"recall"`: The calculated recall value.
  – `"precision"`: The calculated precision value.
  – `"f1_score"`: The calculated F1 score.

• Strictly follow the JSONL format, with one evaluation result per line. Do not output any other content, **only answer in JSONL**, and the model name must be: `merge`.

{"video_id": i, "model": "xxxx", "recall": 0.75,"precision": 0.80, "f1": 0.77}

## C  QALIST

We will show a QAlist for a scene of a video.

---

**QAlist**

**Scene Description:** In a brightly lit, modern room adorned with large windows and lush green plants, a formal interview is underway. Lee Chong Wei, a man in a beige suit, sits across a table from an interviewer in a dark suit. With glasses of water before them, the conversation begins with a profound question. The interviewer asks Lee what he considers success to be. Lee, looking off to the side, thoughtfully responds with Chinese subtitles reflecting his words, stating that by today's standards, he does not feel that he has been successful, setting a contemplative and introspective tone for the discussion.

**Questions and Analysis:**

1. **Temporal:1.1**
   Question: In the interview, did Lee Chong Wei say he doesn't feel successful only after being asked about his view on success?
   **Options:** Yes, No
   **Correct Answer:** Yes

---

**Analysis:** The description clearly states the interviewer asked the question first, and then Lee Chong Wei responded, indicating the correct sequence.

2. **Temporal:1.2**
Question: At the beginning of the interview, did Lee Chong Wei first share his view on success, followed by the interviewer asking questions about it?
**Options:** Yes, No
**Correct Answer:** No
**Analysis:** Incorrect sequence. According to the standard description, the interviewer asked first, and Lee responded afterward.

3. **Temporal:1.3**
Question: In the interview scene, did the act of seating at the table happen before the interviewer posed the profound question about success?
**Options:** Yes, No
**Correct Answer:** Yes
**Analysis:** Correct logic and sequence. The description mentions they "sit across a table" first, then "the conversation begins," indicating seating came before questioning.

4. **Atmosphere**
Question: Was the interview conducted in a dimly lit, cramped room designed to create a tense and oppressive atmosphere?
**Options:** Yes, No
**Correct Answer:** No
**Analysis:** Incorrect environment and atmosphere. The standard description specifies the room was 'brightly lit, modern' with large windows and plants, creating a 'contemplative and introspective' atmosphere, not tense or oppressive.

5. **Emotion**
Question: When asked about success, did Lee Chong Wei demonstrate a thoughtful and introspective state, candidly admitting that by today's standards, he doesn't consider himself successful?
**Options:** Yes, No
**Correct Answer:** Yes
**Analysis:** Accurate depiction of character state. The standard description uses 'thoughtfully responds' and 'contemplative and introspective tone' to portray his state.

6. **Causality**
Question: Did Lee Chong Wei feel unsuccessful because the interviewer provoked him with sharp questions?
**Options:** Yes, No
**Correct Answer:** No
**Analysis:** Incorrect causal relationship. Lee Chong Wei's response stemmed from introspection, not provocation. The interviewer's question is described as 'profound,' not 'sharp.'

7. **Relations**
Question: In the interview, were the two individuals, Lee Chong Wei in a beige suit and the interviewer in a dark suit, portrayed with the formal relationship of interviewer and interviewee?
**Options:** Yes, No
**Correct Answer:** Yes
**Analysis:** Accurate depiction of character relationship. The description clearly mentions 'a formal interview is underway,' establishing their formal dynamics.

8. **Cinematic**
Question: Did the video use close-up shots to capture Lee Chong Wei's teary eyes while answering the question, emphasizing his sadness?

**Options:** Yes, No
**Correct Answer:** No
**Analysis:** Nonexistent element. The standard description only mentions him 'looking off to the side,' with no indication of close-up shots or tears.

9. **Symbolism**
Question: Do the lush green plants in the interview room symbolize that Lee Chong Wei, despite considering himself not successful, still has a lively and vibrant career?
**Options:** Yes, No
**Correct Answer:** Yes
**Analysis:** This is a reasonable symbolic interpretation that connects the environment's details with the theme.

10. **Factuality**
Question: In the interview, was Lee Chong Wei wearing a dark suit while the interviewer wore a beige suit?
**Options:** Yes, No
**Correct Answer:** No
**Analysis:** Factually incorrect. The standard description clearly states Lee Chong Wei wore 'a beige suit,' while the interviewer wore 'a dark suit,' making the question's description the opposite of truth.

## D  MORE DETAILS ABOUT EVALUATION

For closed-source models, we uniformly set the fps to 0.4 for evaluation. For open-source models, except for InternVL3-8B, which has a maximum of 64 frames, all other models use 128 frames. The frames are sampled using a uniform sampling method.

