# OpenReview forum: "LVCap-Eval: Towards Holistic Long Video Caption Evaluation for Multimodal LLMs"
_ICLR.cc/2026/Conference — ICLR 2026 Conference Withdrawn Submission_

### Official Review · Reviewer_mNEN · 2025-10-25

**Soundness:** 3
**Presentation:** 2
**Contribution:** 3
**Rating:** 4
**Confidence:** 3

**Summary:**

This paper introduces LVCap-Eval, a new benchmark for evaluating the long-form video captioning capabilities of Multimodal Large Language Models (MLLMs). Addressing the limitations of existing short-clip benchmarks, LVCap-Eval consists of 200 videos ranging from 2 to 20 minutes across diverse genres. The core contribution is a dual-dimension evaluation protocol that assesses both scene-level narrative coherence and event-level factual accuracy using an LLM-driven pipeline. The authors conduct extensive experiments on 18 MLLMs, revealing a significant performance gap between closed-source models like Gemini-2.5-Pro and their open-source counterparts, especially as video duration increases. The paper also provides a pipeline for generating a training corpus, demonstrating that fine-tuning can substantially improve model performance.

**Strengths:**

1. The work addresses a critical and timely problem: the lack of robust evaluation for long-form video understanding. The proposed dual-dimension evaluation (scene coherence and event factuality) is a significant step beyond existing QA or short-clip captioning benchmarks, offering a more holistic assessment of MLLM capabilities.
2. The benchmark construction is methodologically sound, combining automated tools with human validation to ensure quality and scalability. The experimental evaluation is comprehensive, covering a wide range of state-of-the-art closed-source and open-source models. The further analyses, particularly the experiment isolating the reasoning bottleneck (Table 3), provide valuable insights into the current limitations of MLLMs.
3. The paper is well-written and clearly structured. The annotation and evaluation pipelines are explained effectively, aided by clear diagrams (Figures 1 and 4). The problem statement, methodology, and results are presented in a logical and easy-to-follow manner.

**Weaknesses:**

1. The evaluation protocol is heavily dependent on closed-source models (Gemini-2.5-Pro, GPT-4.1) for key steps like QA generation, event extraction, and final scoring. This raises concerns about reproducibility, potential bias of the judge LLM, and the long-term stability of the benchmark, as the performance of these proprietary models can change over time.
2. The annotation and evaluation pipelines, while semi-automated, require extensive use of powerful, proprietary LLMs, which can be costly and computationally intensive. This may limit the broader community's ability to adopt the benchmark, reproduce results, or extend the dataset.
3. The process of segmenting videos into "scenes" is guided by a prompt to Gemini-2.5-Pro. The definition of a scene can be subjective, and the consistency of this automated segmentation is not quantified. Different LLMs or prompts could yield different segmentations, potentially affecting the comparability of evaluation scores.

**Questions:**

1. The evaluation framework uses GPT-4.1 as a judge. Have you investigated the robustness of this choice? For instance, could you report the agreement rate between different powerful LLMs (e.g., GPT-4.1 vs. Claude 3 Opus) when used as judges? This would help assess whether the reported scores are an artifact of the specific judge model used.
2. The fine-tuned LV-Captioner-Qwen-7B shows impressive gains on LVCap-Eval. Does this improved performance generalize to other long-form video understanding tasks or benchmarks? Evaluating it on an external benchmark would help clarify whether the model has learned a generalizable skill or has simply overfitted to the style and format of your annotation pipeline.
3. Could you provide more details on the consistency of the automated scene segmentation? For example, what is the inter-annotator agreement between human annotators and the LLM-based segmentation, or between different LLMs performing the same segmentation task? This would strengthen the validity of the scene-level evaluation.

---

### Official Review · Reviewer_pehX · 2025-10-25

**Soundness:** 1
**Presentation:** 1
**Contribution:** 2
**Rating:** 2
**Confidence:** 5

**Summary:**

This paper introduces a new benchmark (LVCap-Eval) for evaluating long video captioning. The authors collect 200 long-form videos from YouTube across six domains and propose an evaluation protocol that includes scene-level and fine-grained event-level captions. Videos are segmented into scenes using a shot detection method, and captions are generated for each scene. Additionally, events within scenes are described to provide a more granular level of evaluation.

**Strengths:**

The introduction of a new benchmark for video captioning can benefit the community.

**Weaknesses:**

1.	Misalignment with Long Video Captioning Goals: The benchmark is claimed to target long video captioning. However, the evaluation protocol focuses on scene-level and intra-scene events. This contradicts the core challenge of long video captioning, which involves capturing high-level, temporally extended concepts that span multiple scenes/shuts or the entire video. Therefore, benchmark does not fully address the complexities of long video understanding.

2.	Training Data Assumptions: the training set is also created by this assumption that the events length are limited to the scenes/shots and no longer concepts.

3.	Limited Contribution: The dataset comprises only 200 videos, which is relatively small compared to existing benchmarks. The evaluation results and analysis do not yield strong or novel insights into model architectures or their limitations. For instance, the paper does not explore architectural factors (e.g., attention span, memory mechanisms) that might explain performance differences. Note, the authors do some minor analysis like, frame sampling etc, but they are not significant enough.

4.	Comparative Analysis Gaps: Table 1 shows that several existing datasets contain longer videos than the proposed benchmark. A fair comparison would require building upon or extending those datasets, accompanied by a qualitative analysis to demonstrate how the new benchmark offers distinct advantages.

5.	Lack of any Qualitative Comparison: The paper lacks a section that qualitatively compares the proposed benchmark with existing ones. Furthermore, while Table 1 provides some quantitative comparisons, it omits important statistics such as the number of videos.

6.	Inadequate Related Work Discussion: The related work section on video captioning includes references to image captioning benchmarks (e.g., Dong et al., 2024; Lu et al., 2025), which are not directly comparable. The claim that these image-based benchmarks are limited to short clips and that the proposed benchmark overcomes these limitations is misleading.

7.	Lack of Caption Density Statistics: The authors claim their captions are "dense," but provide no quantitative evidence to support this. Metrics such as the average number of captions per video would help substantiate this claim.

8.	Presentation Issues: The paper presentation needs fundumental improvement to become suitable for a top-conference like ICLR. For example:
- The paper is about introducing a new benchmark, but they do not include even one sample of the benchmark in the paper. There are a couple of vague examples in some figures like Figure 1, but it is not even cleared that if they are from the LVCap-Eval benchmark.
- At the end of the Introduction, they talk that thier generation pipline is one of their contribution while they did not mention it earlier in the intro.
- They have Table 1 where, they qualitatively compare the current datasets. Howvere, they do not even refer to this comparison in text.

**Questions:**

Please check the weaknesses section

---

### Official Review · Reviewer_6RWE · 2025-10-28

**Soundness:** 2
**Presentation:** 2
**Contribution:** 2
**Rating:** 2
**Confidence:** 4

**Summary:**

This paper presents LVCap-Eval, a comprehensive benchmark for evaluating long-form video captioning in MLLMs. To promote model improvement, the authors build a 7,000-video training corpus using an automated hybrid annotation pipeline. Extensive experiments on 18 state-of-the-art models reveal that closed-source systems like Gemini-2.5-Pro significantly outperform open-source ones, whose performance declines with increasing video length. Further analyses show that the main limitation of current MLLMs lies in high-level narrative reasoning rather than visual perception, and that simple context-aware or memory-based mechanisms can effectively mitigate long-range dependency challenges.

**Strengths:**

The paper makes a contribution by introducing a long-form video captioning benchmark that jointly evaluates narrative coherence and factual accuracy, offering both methodological novelty and strong empirical validation. The analysis and experiments provide valuable insights and resources for advancing multimodal large language models in extended-duration video understanding.

**Weaknesses:**

1. It is unclear whether the scene-level evaluation verifies the chronological order of events.

2. Although the appendix explains the frame extraction strategy, it does not specify the resolution used. As far as I know, different models adopt different default resolutions.

3. What is the impact of using models with different input sizes?

4. How is the accuracy of the training corpus verified?

6. How does caption length vary with video duration?

7. The paper’s presentation, especially the figures, needs improvement. For example, the text in Figure 2 is too dense, Figure 3’s font is too small, and the row heights in Table 2 look inconsistent.

8. The scene-level evaluation method appears almost identical to VDC, weakening the novelty and contribution.

9. When comparing GPT-4-mini with other GPT variants, does it cause hallucinations or information leakage? How is the reliability of training data ensured?

10. The paper only tests one model configuration for input frame sampling; conclusions may lack generality.

11. Sampling up to 256 frames for a 20-minute video seems clearly unreasonable (even with the 128-frame “optimal” claim), implying about one frame every five seconds. If sampling by FPS rather than uniform intervals, what would the results look like?

13. The paper lacks necessary statistical reporting, such as average caption length.

**Questions:**

See weakness.

---

### Official Review · Reviewer_D2kd · 2025-11-01

**Soundness:** 3
**Presentation:** 3
**Contribution:** 2
**Rating:** 6
**Confidence:** 4

**Summary:**

This paper highlights that current MLLM evaluations focus only on short clips, neglecting the predominance of long videos in real-world scenarios. To address this gap, the authors benchmark the captioning ability of current MLLMs on extremely long videos and propose LVCap-Eval, a holistic evaluation framework. The benchmark comprises 200 long videos across six domains for testing. For comprehensive evaluation both at scene-level and event-level, it introduces a dual-dimension evaluation protocol with distinct tasks for each level. The study validates the benchmark's effectiveness through extensive experiments on 18 state-of-the-art MLLMs.

**Strengths:**

- The paper addresses a crucial gap by creating a benchmark for extremely long videos for MLLMs, a field previously neglected. This is a meaningful task because most real-world application scenarios for MLLM captioning involve long videos.
- The authors construct a dataset of 200 long videos with precise labels. This effort involved significant work and represents a substantial contribution to the research community.
- The design of the evaluation framework is appropriate and holistic. The paper assesses scene-level understanding through Scene QA tasks and event-level memory through narrative tasks. The evaluation is end-to-end, and scoring is conducted by off-the-shelf MLLMs, making the protocol easy to follow and implement.

**Weaknesses:**

- The paper is not very insightful. The main conclusion from comparisons is that closed-source models beat open-source models on extremely long video captioning, which is not surprising. It is recommended that the authors provide more detailed analysis of the comparison results, especially on failure of open-sourced models, to offer researchers greater insight into specific areas for improvement, thereby guiding the advancement of open-source models.
- Some details regarding the evaluation protocol require further clarification.
- The paper uses Gemini2.5-pro as the grader. Have the authors investigated the potential for induced bias within this evaluation model? The evaluation would be more convincing if authors could provide comparisons with other scoring models. Futhermore, could the authors present a comparison between MLLMs and human scoring to demonstrate there is no non-negligible gap? (i.e., human-aligment issues)

**Questions:**

- Figure 4 illustrates a cross-check between Ground Truth (GT) events and generated events to compute captioning accuracy. Could the authors clarify how this cross-check is performed? Given that events are expressed in natural language, their formats will inevitably vary. Is an LLM utilized here to cross-check the matches of the subject, action, and key entities?

---

### Note · Authors · 2025-11-12

I have read and agree with the venue's withdrawal policy on behalf of myself and my co-authors.